# Dynamic modulation of social gaze by sex and familiarity in marmoset dyads

Feng Xing[1,2], Alec G Sheffield[1,2,3], Monika P Jadi[2,3,4†], Steve WC Chang[2,4,5,6†], Anirvan S Nandy[2,4,5,6*†]

[1]Inderdepartmental Neuroscience Program, Yale University, New Haven, United States; [2]Department of Neuroscience, Yale University, New Haven, United States; [3]Department of Psychiatry, Yale University, New Haven, United States; [4]Wu Tsai Institute, Yale University, New Haven, United States; [5]Department of Psychology, Yale University, New Haven, United States; [6]Kavli Institute for Neuroscience, Yale University, New Haven, United States

## eLife Assessment

This study establishes a methodology (machine vision and gaze pose estimation) and behavioral apparatus for examining social interactions between pairs of marmoset monkeys. It has been difficult to study social interactions using artificial stimuli rather than genuine interactions between unrestrained animals. This study makes a **fundamental** contribution to social neuroscience research in a laboratory setting. Their results are **convincing** showing that the study of unrestrained social interactions is possible with detailed quantification of position and gaze. The methodology presented here is relevant to research in social neuroscience, neuroethology, and primatology.

**\*For correspondence:**
anirvan.nandy@yale.edu

†These authors contributed equally to this work

**Competing interest:** The authors declare that no competing interests exist.

**Abstract** Social communication relies on the ability to perceive and interpret the direction of others' attention, and is commonly conveyed through head orientation and gaze direction in humans and nonhuman primates. However, traditional social gaze experiments in nonhuman primates require restraining head movements, significantly limiting their natural behavioral repertoire. Here, we developed a novel framework for accurately tracking facial features and three-dimensional (3D) head gaze directions of multiple freely moving common marmosets (*Callithrix jacchus*). By combining deep learning-based computer vision tools with triangulation algorithms, we were able to track the facial features of marmoset dyads within an arena. This method effectively generates dynamic 3D geometrical facial frames while overcoming common challenges like occlusion. To detect the head gaze direction, we constructed a virtual cone, oriented perpendicular to the facial frame. Using this pipeline, we quantified different types of interactive social gaze events, including partner-directed gaze and joint gaze to a shared spatial location. We observed clear effects of sex and familiarity on both interpersonal distance and gaze dynamics in marmoset dyads. Unfamiliar pairs exhibited more stereotyped patterns of arena occupancy, more sustained levels of social gaze across social distance, and increased social gaze monitoring. On the other hand, familiar pairs exhibited higher levels of joint gazes. Moreover, males displayed significantly elevated levels of gazes toward females' faces and the surrounding regions, irrespective of familiarity. Our study reveals the importance of two key social factors in driving the gaze behaviors of a prosocial primate species and lays the groundwork for a rigorous quantification of primate behaviors in naturalistic settings.

## Introduction

Primates, including humans, exhibit complex social structures and engage in rich interactions with members of their species, which are crucial for their survival and development. Among social stimuli, the face holds paramount importance with specialized neural systems (*Deen et al., 2023*; *Hesse and Tsao, 2020*) and is attentively prioritized by primates during much of their social interaction. Notably, the eyes garner the most attention among all facial features, playing a pivotal role in indicating the direction of others' attention and also possibly their intention (*Dal Monte et al., 2015*; *Emery, 2000*; *Itier et al., 2007*). Indeed, understanding and interpreting the gaze of fellow individuals is a fundamental attribute of the theory of mind (*Martin and Santos, 2016*; *Saxe and Kanwisher, 2003*). While current studies of social gaze using either realistic stimuli or pairs of rhesus macaques (*Macaca mulatta*) in a controlled laboratory setting (*Dal Monte et al., 2016*; *Mosher et al., 2014*; *Ramezanpour and Thier, 2020*; *Shepherd et al., 2006*; *Shepherd and Freiwald, 2018*) provide valuable insights into social gaze behaviors, they are nevertheless limited in their ecological relevance. Many laboratory paradigms rely on head-restrained animals and simplified stimuli, decoupling eye movements from natural head-body dynamics and limiting spontaneous social behavior (*Land, 2006*; *Einhäuser et al., 2007*; *Foulsham et al., 2011*). Social gaze is also typically examined in tightly controlled dyadic or screen-based paradigms, which lack the reciprocal, context-dependent, and multi-agent dynamics of natural social interactions (*Shepherd, 2010*; *Birmingham et al., 2008*). In contrast, gaze behavior in natural settings is shaped by locomotion, body posture, spatial relationships, and social structure – factors largely absent from laboratory studies (*Emery, 2000*; *Klein et al., 2009*). Consequently, the generalizability of findings from controlled experiments to freely moving, socially embedded interactions remains unclear, motivating the need for more naturalistic yet quantitative approaches.

To address these limitations, we turned to common marmosets (*Callithrix jacchus*), a highly prosocial primate species known for their social behavioral and cognitive similarities to humans (*Miller et al., 2016*). Marmosets are also a model system with increasing applications in computational ethology (*Mitchell et al., 2014*; *Ngo et al., 2022*). Like humans, they engage in cooperative breeding, a social system in which individuals care for offspring other than their own, usually at the expense of their reproduction (*French, 1997*; *Solomon and French, 1997*). Gaze directions, inferred from head orientation, hold crucial information about marmoset social interactions (*Heiney and Blazquez, 2011*; *Spadacenta et al., 2022*). The emergence of computational ethology (*Anderson and Perona, 2014*; *Datta et al., 2019*) has propelled the development of a host of computer vision tools using deep neural networks (e.g. OpenPose by *Cao et al., 2017*, DeepLabCut by *Mathis et al., 2018*, DANNCE by *Dunn et al., 2021*, SLEAP by *Pereira et al., 2022*). Moreover, we have recently developed a scalable marmoset apparatus for automated pulling (MarmoAAP) for studying cooperation in marmoset dyads (*Meisner et al., 2024*). However, tracking head gaze direction in multiple freely moving marmosets poses a challenging problem, not yet solved by existing computer vision tools. This problem is further complicated by the fact that accurately tracking gaze directions in primates requires three-dimensional (3D) information.

Here, we propose a novel framework based on a modified DeepLabCut pipeline, capable of accurately detecting body parts of multiple marmosets in 2D space and triangulating them in 3D space. By reconstructing the face frame with six facial points in 3D space, we can infer the head gaze of each animal across time. Importantly, marmosets that are not head-restrained use rapid head movements for reorienting and visual exploration (*Pandey et al., 2020*), and therefore the head direction serves as an excellent proxy for gaze direction in unrestrained marmosets. With this framework in place, we investigated the gaze behaviors of male-female pairs of freely moving and interacting marmosets to quantify their social gaze dynamics. We investigated these gaze dynamics along the dimensions of sex and familiarity and found several key differences along both of these important social dimensions, including increased partner gaze in males that is modulated by familiarity, increased joint gaze among familiar pairs, and increased gaze monitoring by males. This fully automated tracking system can thus serve as a powerful tool for investigating primate group dynamics in naturalistic environments.

## Results

### Experimental setup and reconstruction of video images in 3D space

The experimental setup consisted of two arenas made with acrylic plates that allowed two marmosets to visually interact with each other while being physically separate (*Figure 1—figure supplement 1A*). Each arena was 60.96 cm long, 30.48 cm wide, and 30.48 cm high. Five sides of the arena, except the bottom, were transparent, allowing a clear view of the animal subjects under observation. The bottom side of the arena was perforated with 1-in diameter holes arranged in a hexagonal pattern to aid the animal's traction. The arenas were mounted on a rigid frame made of aluminum building blocks, with the smaller sides facing each other, and were separated by a distance of 30.48 cm. A movable opaque divider was placed between the arenas during intermittent breaks to prevent the animals from having visual access to each other (Methods). Two monitors were attached to the aluminum frame, one on each end, for displaying video or image stimuli to the animals. To capture the whole experimental setup, two sets of four GoPro 8 cameras were attached to the frame, where each set of cameras captured the view of one of the arenas.

After obtaining the intrinsic parameters of the cameras by calibration and the extrinsic parameters of the cameras by L-frame analysis (see Methods), we established a world coordinate system of each arena surrounded by the corresponding set of four cameras. Crucially, the two independent world coordinate systems of the two arenas were combined by measuring the distance between the two L-shaped frames and adding this offset to one of the world coordinate systems.

With the established world coordinate system, any point captured by two or more cameras could be triangulated into a common 3D space. Thus, the experimental setup was reconstructed into 3D space by manually labeling the vertices of the arenas and monitors in the image space captured by the cameras (*Figure 1—figure supplement 1B*). The cameras on the monitor ends (marked as 'ML1', 'ML2', 'MR1', 'MR2' in *Figure 1—figure supplement 1B*) recorded both animal subjects, whereas the cameras in the middle (marked as 'OL1', 'OL2', 'OR1', 'OR2' in *Figure 1—figure supplement 1B*) recorded only one animal subject.

### Automatic detection of facial features of two marmosets

Six facial features – the two tufts, the central blaze, two eyes, and the mouth (*Figure 1A*) – were selected for automated tracking using a modified version of a deep neural network (DeepLabCut, *Mathis et al., 2018*). The raw video from each camera was fed into the network to compute the probability heatmaps of each facial feature. We modified the original method to detect features of two animals (*Figure 1B*). After processing the raw video, two locations with the highest probability over a threshold (95%) were picked from each probability heatmap (*Figure 1B*, *feature detection*). Since all the features from the same animal should be clustered in image space, a K-means clustering algorithm (*Figure 1B*, *initial clustering*) was used on the candidate features with the constraint that one animal can only have one unique feature (*Figure 1B*, *refine clustering*); e.g., one animal cannot have two left eyes. After clustering, two clusters of features corresponding to the two animals were obtained. To detect outliers that were not valid features, we first calculated a distribution of within-cluster distances (*Figure 1B*, *remove outliers*). Outliers were determined as those points that had nearest-neighbor distances that were two standard deviations above the average within-cluster distance and were excluded from subsequent analyses. Note that the above analyses were performed independently for each video frame.

To establish temporal continuity across video frames and track animal identities, we first calculated the centroid of each cluster (*Figure 1B*, *calculate centroids*). Under the heuristic that centroid trajectories corresponding to each individual animal are smoothly continuous in space and time (i.e. there are no sudden jumps or reversals in centroid location across frames at our sampling rate of 30 Hz), we assigned identities to the centroids, thus enabling us to track identities over time (*Figure 1B*, *establish identities*). The facial features corresponding to a centroid inherited this identity (*Figure 1B*, *cluster with identities*). We tested our automated pipeline against a manually labeled dataset and found that features were detected with high precision (root mean squared error = 3.31 ± 0.1 pixels [mean ± s.e.m.]; *n*=2400 facial features; see Methods).

Our method thus allowed us to accurately detect and track the facial features of two marmosets in the arena (*Figure 1C*), and in more general contexts, such as in a home cage with occlusions (*Video 1*).

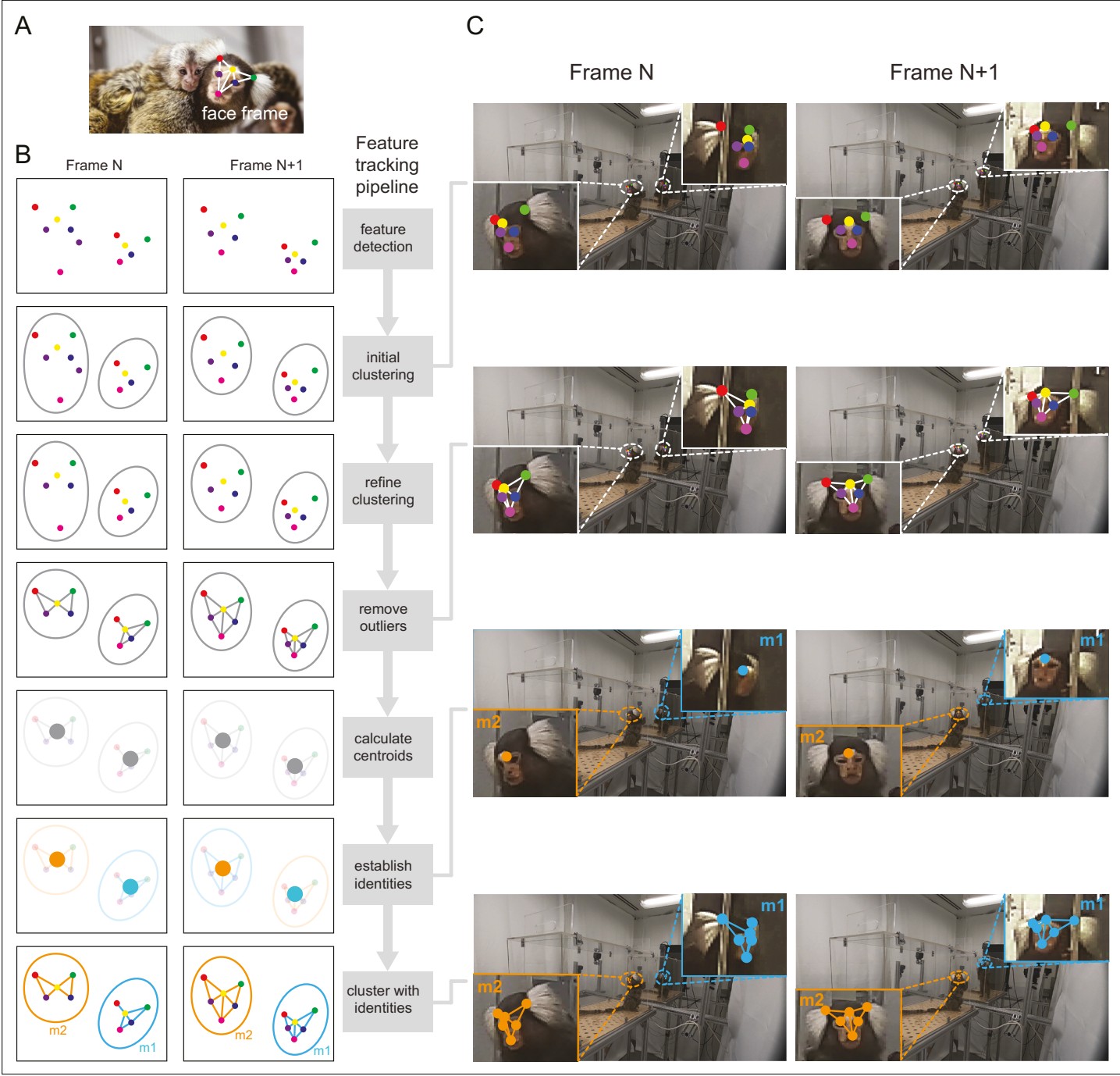

**Figure 1.** Pipeline of detecting facial features of two marmosets. (**A**) Six facial features of the marmoset (face frame) are color-coded: right tuft (red), central blaze (yellow), left tuft (green), right eye (purple), left eye (blue), and mouth (magenta). (**B**) Feature tracking pipeline (right) with the corresponding illustration for each step across two adjacent video frames (left). At the end of the pipeline, the facial features are clustered with the identities assigned consistently across frames. Facial points are color-coded as in A. (**C**) Example frames of four steps in the pipeline shown in B. It can be seen clearly that the facial points are tracked and clustered accurately, and the identities are consistent across frames.

The online version of this article includes the following figure supplement(s) for figure 1:

**Figure supplement 1.** Experimental setup and reconstruction in three-dimensional (3D) space.

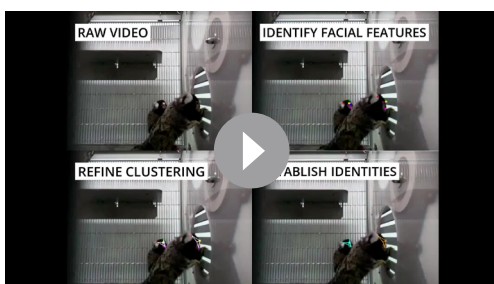

**Video 1.** Results from different processing stages of the facial features detection pipeline are shown for two marmosets in their home cage.
https://elifesciences.org/articles/105034/figures#video1

## Inferring head gaze direction in 3D space

In our experimental setup, the cameras in the middle unambiguously recorded only one animal. The centroids of the facial features in the image space recorded by the middle cameras were triangulated into 3D space and were constrained to be confined within the bounds of the arena. Any missing centroids were filled by interpolation from neighboring frames from both the past and future time points. The triangulated centroids acquired from cameras in the middle were then projected into the image space of cameras on the monitor ends. Since both animals were recorded by the cameras on the monitor ends, facial features from these camera views were detected with identities assigned (as described in the 2D pipeline). In the image space of cameras on the monitor ends, the cluster of facial feature points closer to the projected centroid and within the bounds of the arena was kept for later triangulation.

The detected facial features captured by all four cameras for one animal were subjected to triangulation. For each feature, results from all possible pairs of four cameras were triangulated. All the triangulation results were averaged to yield the final coordinates of the body part in 3D space. Any missing features were filled by interpolation from neighboring frames, including the previous and future time points. Testing against a manually labeled and triangulated dataset yielded a high precision of 3D feature detection (root mean squared error = 3.7 ± 0.1 mm [mean ± s.e.m.]; $n$=2400 facial features; see Methods).

The six facial points constituted a semi-rigid geometric frame as the animal moves in 3D space ('face frame'; *Figure 1*), allowing us to infer the animal's head gaze direction as follows. The gaze direction was calculated as the normal to the facial plane ('face norm vector') defined by the two eyes and the central blaze. The position of the ear tufts, which were behind this facial plane, was used to determine gaze direction. Since marmoset saccade amplitudes are largely restricted to 10 degrees (median less than 4 degrees) (*Mitchell et al., 2014*), we modeled the head gaze as a virtual cone with a solid angle of 10 degrees ('gaze cone') emanating from the facial plane (*Figure 2A*). Notably, with multiple camera views, the face frame can be reconstructed even when the face was invisible to one of the cameras, such that the reconstructed face frame in 3D can be projected back into the image space to validate the accuracy of the detection and reconstruction (*Figure 2B*). We were thus able to obtain the animal's continuous movement trajectory and the corresponding gaze direction over time (*Figure 2C*, *Video 2*). To characterize stable gaze epochs for subsequent analyses, we calculated the velocity of the face norm vector and applied a stringent threshold of 0.05 (normalized units) below which the marmoset's head movement was considered stationary (*Figure 2D*).

We first examined our method's ability to characterize gaze behaviors of freely moving marmosets by presenting either video or image stimuli to individual animals on a monitor screen (see Methods). We analyzed gaze behavior following the onset of video stimuli presented at different monitor locations (*Figure 2—figure supplement 1A*). The analysis was restricted to video clips in which human annotators confirmed that the marmosets were looking at the monitor. Consistent with prior work in head-fixed marmosets (*Mitchell et al., 2014*), gaze-monitor intersection centers clustered around the corresponding stimulus locations after stimulus onset, supporting the spatial accuracy of the estimated gaze directions. Marmosets exhibited longer gaze duration to video stimuli compared to image stimuli (*Figure 2—figure supplement 1B*; Mann-Whitney U test, p<0.001). However, this difference was not caused by differences in gaze dispersion (*Figure 2—figure supplement 1C*; Mann-Whitney U test, ns). By examining the frequency of gaze events, we found that marmosets gazed at the monitor more during the early period of video stimuli presentations compared to the late period (*Figure 2—figure supplement 1D*; Mann-Whitney U test, p<0.001), while there was no such difference for the image stimuli (*Figure 2—figure supplement 1D*; Mann-Whitney U test, p=0.4345). There was also a significant difference in gaze frequency between the early period of video presentations compared to

**Figure 2.** Three-dimensional (3D) reconstruction of facial features and head gaze modeling. (**A**) The face frames of two marmosets are reconstructed in 3D using the tracked facial points in *Figure 1A*. A cone perpendicular to the face frame (gaze cone; 10-degree solid angle) is modeled as the head gaze. (**B**) Two example frames with the facial points projected from 3D space onto different camera views are shown. The left frame demonstrates that the facial points can be detected using information from other cameras, even if the face is invisible from that viewpoint. (**C**) Trajectory of the reconstructed face frame and the corresponding gaze cones across time. (**D**) The histogram of the face norm velocity. The red dotted line (0.05) is the threshold below which the marmoset head direction is considered to be stationary.

The online version of this article includes the following figure supplement(s) for figure 2:

**Figure supplement 1.** Gaze behavior analysis of a single marmoset viewing stimuli on the monitor.

the same period for image presentations (*Figure 2—figure supplement 1D*; Mann-Whitney U test, p<10$^{-10}$). Taken together, our results support that dynamic visual stimuli elicit greater overt attention compared to static stimuli in marmosets, similar to macaques (*Dal Monte et al., 2016*; *Furl et al., 2012*) and humans (*Chevallier et al., 2015*).

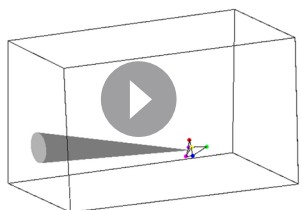

**Video 2.** Three-dimensional (3D) reconstruction of the face frame and the inferred gaze cone across time for a single marmoset.

https://elifesciences.org/articles/105034/figures#video2

## Positional dynamics of marmoset dyads

With this automated system in place, we recorded the behavior of four pairs of familiar marmosets and four pairs of unfamiliar marmosets. We first observed that preferred spatial position was highly non-uniform, with animals preferring to occupy the ends of the elongated arena rather than the center. Data from each pair was recorded in one session consisting of ten 5 min free-viewing blocks interleaved with 5 min breaks. Each pair consisted of a male and a female animal. The familiar pairs were defined as cage mates, while

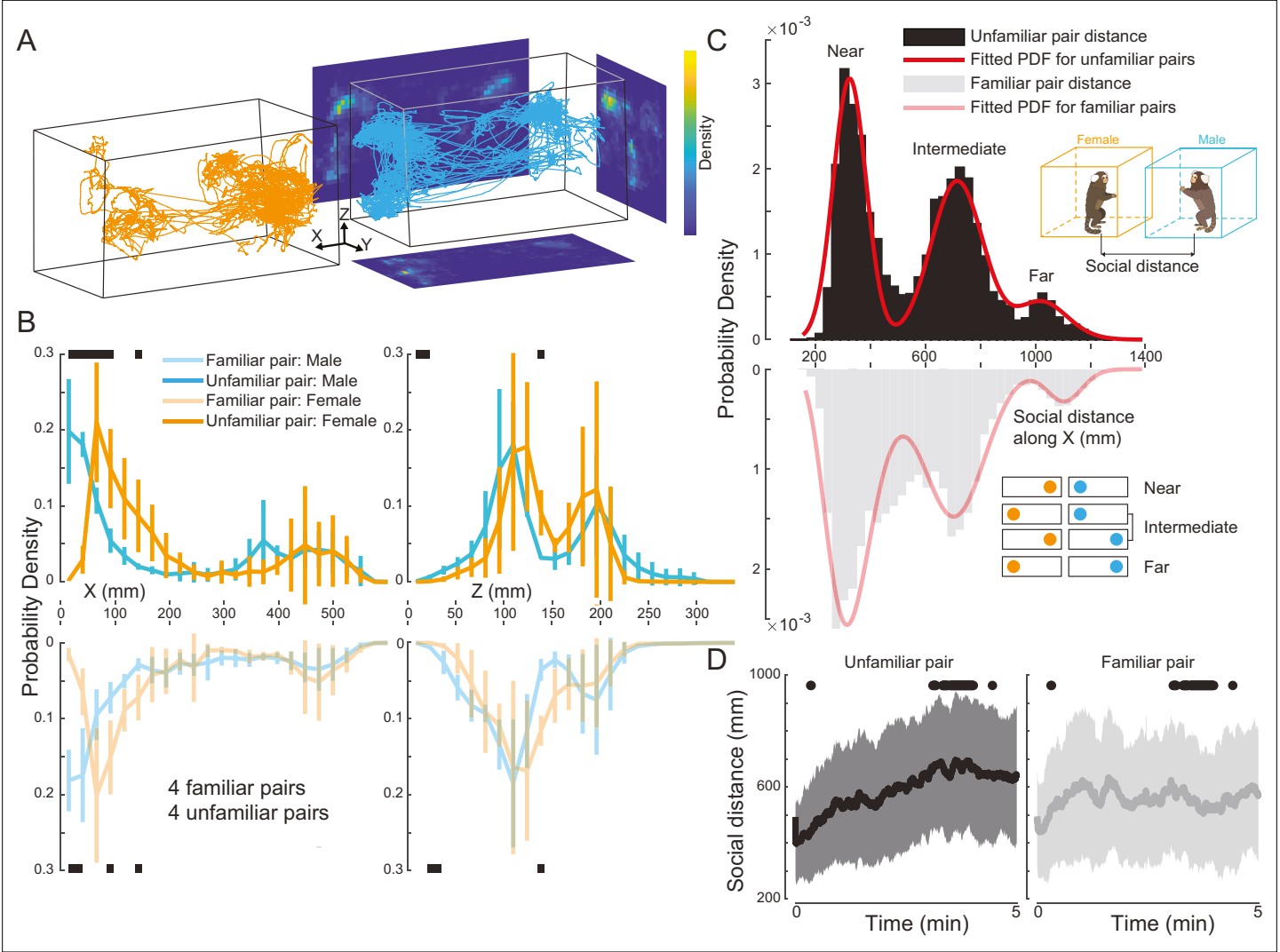

**Figure 3.** Positional dynamics of marmoset dyads. (**A**) Movement trajectories of the face frame centroids for a marmoset pair (orange for female, cyan for male) in an example 5 min block. The heatmaps were calculated using the projections of the trajectories to *XY*, *YZ*, and *XZ* planes. (**B**) Marginal distributions of movement trajectories along the *X* and *Z* axes were calculated for all marmosets and grouped by familiarity and sex (transparent colors for familiar pairs, opaque colors for unfamiliar pairs). Black bars indicate significant differences between pairs of distributions (Mann-Whitney U test, significance level at 5%). (**C**) Histograms of social distance along the *X* axis show trimodal distributions for both familiar pairs (gray) and unfamiliar pairs (black). The top inset shows a schematic of the arena configuration with corresponding colors (orange for females and cyan for males). Social distance is defined as the Euclidean distance between the two marmosets. The fitted red curve for the unfamiliar pairs is a tri-Gaussian distribution, while the fitted red curve for the familiar pairs is a mixture of Gamma and Gaussian distributions, with the first peak as the Gamma distribution. The three regions were designated as 'Near', 'Intermediate', and 'Far'. The bottom inset illustrates the reason for this nomenclature. (**D**) Temporal evolution of social distance for unfamiliar and familiar pairs (with each 5 min viewing block). The central dark line is the mean, and the shaded area is the standard deviation. Black dots indicate significant differences (Mann-Whitney U test, significance level at 5%).

each member of an unfamiliar pair was from a different home cage with no visual access to each other while in the colony. We first examined the movement trajectories of marmoset dyads and used the centroids of the face frames across time to represent the trajectories. For an example 5 min segment (*Figure 3A*), we observed that the marmosets preferred to stay at the two ends of their respective arenas. This was confirmed by a heatmap of projections of the trajectories on the plane parallel to the vertical long side of the arenas ('*XZ*' plane). Furthermore, there were two hotspots along the vertical axis in the heatmap of projections to the vertical short side plane ('*YZ*' plane), suggesting that the animals' preferred body postures were either upright or crouched.

Second, we found a distinct sexual dimorphism in spatial positioning, where males consistently stayed closer to their partners than females. We examined the marginal distributions of the

movement trajectories along the horizontal ('*X*') and vertical ('*Z*') axes across all sessions and grouped them along the dimensions of sex and familiarity (*Figure 3B*). Along the *X* axis (*Figure 3B*, left), the distributions were slightly bimodal, with the main peak in the region near to the inner edge of the arenas. Regardless of familiarity, male marmosets tended to stay closer to the inner edge compared to females, as shown by the significant differences in the distributions when *X* ranged from 0 to 150 mm. However, there were no significant differences between the same-sex members of familiar and unfamiliar pairs. For the *Z* axis (*Figure 3B*, right), the distributions were bimodal (bimodality coefficients [BC] all exceeded the threshold at 5/9 with a 5% margin; BC(familiar male)=0.5863; BC(familiar female)=0.6278; BC(unfamiliar male)=0.6235; BC(unfamiliar female)=0.6570; Warren Sarle's bimodality test), consistent with what we observed in the heatmaps, indicating either upright or crouched postures. The positional distributions along the *Z* axis were not different based on sex or familiarity.

Third, our analysis of dyadic distance revealed that familiar pairs spend more time in close proximity and maintain more flexible social distances over time compared to unfamiliar pairs. To characterize the social distance dynamics of the freely moving dyads, we calculated the distance between the centroids of the pairs. We then examined the distributions of the social distance along the *X* axis separately for familiar and unfamiliar pairs (*Figure 3C*). The distributions were trimodal and can be explained by the bimodal distribution of movement trajectories of individual marmosets along the *X* axis. As mentioned above, marmosets tended to stay at the two ends of their arenas, and thus, combinations of preferred positions at the two ends for the dyads (see insets in *Figure 3C*) resulted in the trimodal distribution. We termed these three peaks as 'Near', 'Intermediate', and 'Far'. To quantify these distributions, we fitted the empirical data with mixture models using maximum likelihood estimation (see Methods). The social distance for unfamiliar pairs was best fitted by a mixture of three Gaussians, while the distribution for familiar pairs was best fitted by a mixture of Gamma and Gaussian distributions. The first peak ('Near') of the familiar-pair distribution was best fitted by a Gamma distribution, implying a higher degree of dispersion when familiar marmosets were close to each other. Upon examining the temporal evolution of the social distance (within each 5 min viewing block), we found that the social distance of unfamiliar pairs increased over time, whereas this distance fluctuated over time for familiar pairs (*Figure 3D*), further indicating that the social distance dynamics of marmoset dyads depended on familiarity.

## Social gaze dynamics of marmoset dyads

We next investigated the interactive aspects of gaze behaviors in freely moving marmoset dyads. We found that social interaction in marmosets is characterized by a distinct sexual dimorphism in gaze interest and a transition from 'monitoring' in unfamiliar pairs to 'active reciprocation' in familiar pairs. The gaze interaction between two animals could be simplified as the relative positions of two gaze cones in 3D space (see Methods; *Figure 4A*; *Video 3*). If the gaze cone of one animal intersected with the facial plane of the second animal (but not vice versa), we termed it 'partner gaze'. If the gaze cones of both animals intersected with that of the other's facial plane, we termed it 'reciprocal gaze'. In our dataset, the instances of reciprocal gaze were very low and were thus excluded from further analysis. If the two cones intersected anywhere outside the facial planes, we termed it 'joint gaze'. All other cases were regarded as 'no interaction' between the two animals.

Our analysis of gaze states reveals that male marmosets maintain a high baseline of interest in females, though this relationship depends on familiarity and spatial proximity. We identified stable gaze epochs by thresholding the face norm vector velocity (*Figure 2D*). Stable epochs were categorized into gaze states based on the gaze event types described above. We first analyzed the fraction of gaze states in the three position ranges identified from the social distance analysis (*Figure 4B and C*). We found that male marmosets gazed more toward their partner females' faces regardless of familiarity (p<0.01, $\chi^2$ test). However, while male interest in familiar females decreased as social distance increased, interest in unfamiliar females remained constant across different social distances, suggesting a persistent monitoring of novel partners (*Figure 4C*). Moreover, females in unfamiliar pairs exhibited significantly higher partner gazes (female→male) compared to those in familiar pairs (p<0.01, $\chi^2$ test; *Figure 4C*). The total counts of social gaze states (joint gaze and partner gaze) were higher for familiar pairs when they were near, but these decreased more dramatically with increasing distance (*Figure 4B*).

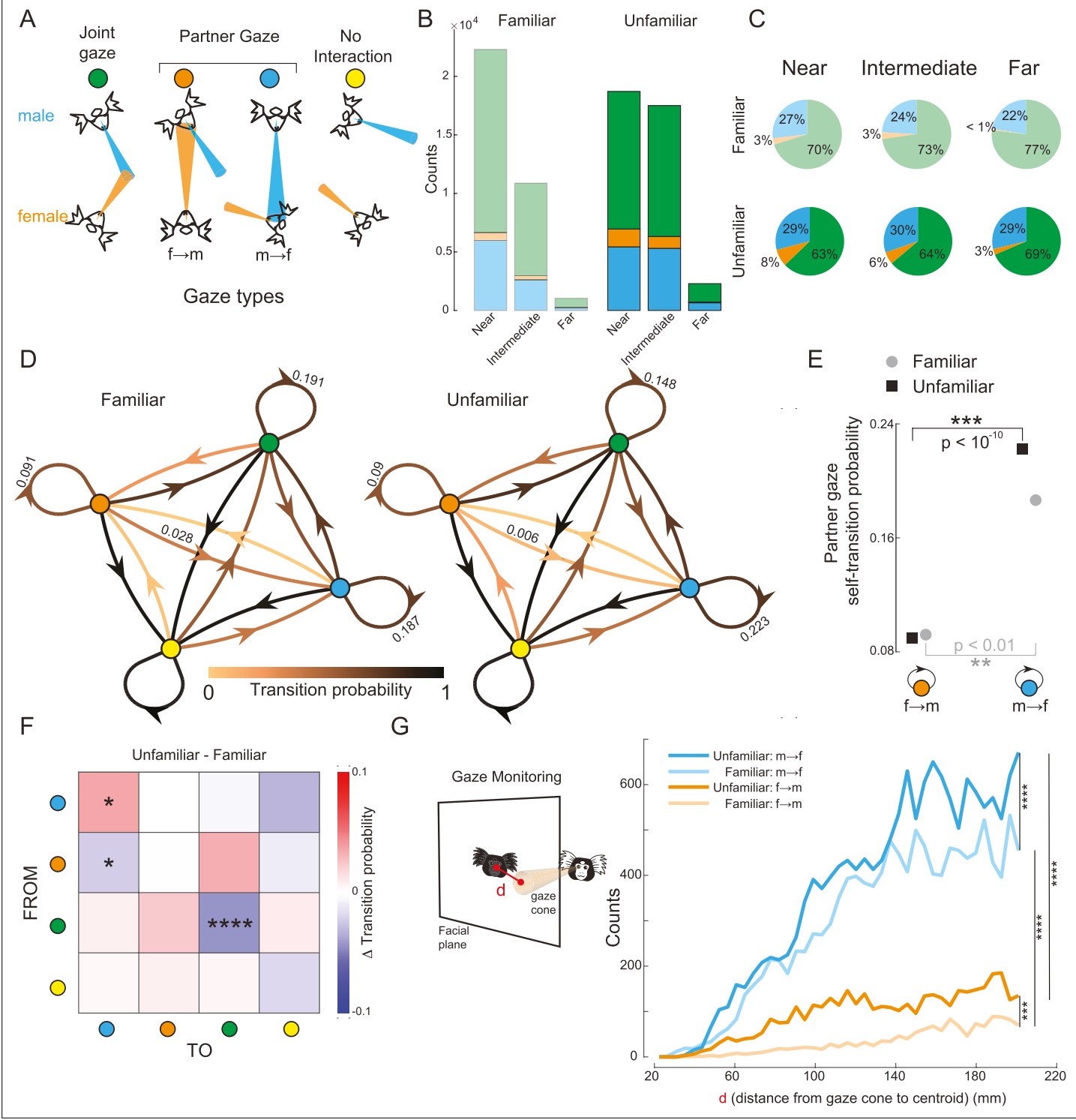

**Figure 4.** Live interactive gaze analysis of unfamiliar and familiar marmoset dyads. (**A**) Gaze type categorized based on the relative positions of the gaze cones. Joint gaze is defined as two marmosets looking at the same location. A partner gaze is defined as one animal looking at the other animal's face (but not vice versa). No interaction occurs when the two gaze cones do not intersect. (**B**) Histograms of gaze count as a function of inter-animal distance, shown separately for familiar and unfamiliar pairs. (**C**) Same data as in B shown as pie charts of percentages in social gaze states. (**D**) Gaze state transition diagrams for familiar and unfamiliar pairs. The nodes are the gaze states, and the edges connecting the nodes represent the transition between states. Edge colors indicate transition probabilities. (**E**) Partner gaze self-transition probabilities for familiar and unfamiliar pairs ($\chi^2$ test). (**F**) Delta transition matrix between the unfamiliar pair and familiar pair state transition diagrams. Transitions that are significantly different across familiarity are marked by asterisks ($\chi^2$ test, male to male, p<0.05; female to male, p<0.05; joint to joint, p<0.0001). (**G**) Left, The schematic illustrates how

*Figure 4 continued on next page*

*Figure 4 continued*

gazing toward the surrounding region of a partner's face area was measured. Right, Counts of gaze toward the surrounding region of the partner's face by familiarity and sex (Mann-Whitney U test, *** means p<0.001; **** means p<0.0001).

Our state transition analysis revealed that gaze dynamics are governed by a combination of persistent male attention and a shared tendency for males to follow the female's gaze into joint attention. To investigate the dynamics of these gaze states, we computed state transition probabilities among distinct gaze event types for familiar and unfamiliar dyads. Markov chain modeling of state transitions further highlighted that familiar dyads engage in more complex, reciprocal social exchanges compared to the one-sided monitoring seen in unfamiliar dyads (*Figure 4D*). We first focused on the recurrent (self-transition) edges for the partner gaze states. Recurrent edges indicate a transition back to the same stable gaze state after a break, likely due to physical movement, and reflect the robustness of the state despite movement. In line with our previous results (*Figure 4B and C*), males exhibit significantly higher recurrent partner gazes compared to females, irrespective of familiarity ($\chi^2$ test, unfamiliar male vs unfamiliar female, $p<10^{-10}$; familiar male vs familiar female, p<0.01; *Figure 4E*). Furthermore, we observed a consistent pattern of gaze following across both groups; approximately 17% of male-to-female partner gazes transitioned directly into joint gaze (*Figure 4D*), suggesting that males frequently utilize the female's gaze orientation to guide their own environmental exploration.

Our comparison of state transition probabilities reveals that familiarity transforms male gaze behavior from a persistent monitoring of novel partners to a more socially responsive, reciprocal interaction (*Figure 4F*). First, recurrent male partner gaze (male→female) was significantly enhanced in unfamiliar pairs (p<0.05, $\chi^2$ test), suggesting a heightened interest in unfamiliar females. Second, there was a higher probability of transition from a female partner gaze to a male partner gaze in familiar pairs compared to unfamiliar pairs, suggesting that familiar males have a greater awareness of and tendency to reciprocate their partners' gaze (p<0.05, $\chi^2$ test). Third, there was a higher probability of recurrent joint gazes in familiar pairs compared to unfamiliar pairs, suggesting that familiar pairs explore common objects more than unfamiliar pairs ($p<10^{-4}$, $\chi^2$ test).

Monitoring others to anticipate their future actions is critical for successful social interactions (*Hari et al., 2015*). In particular, successful interactive gaze exchanges require constant monitoring of others' gaze. In addition to direct eye contact, we found that unfamiliar marmosets engage in increased peripheral monitoring of their partners. We analyzed the gaze distribution in the surrounding region of a partner's face to estimate gaze monitoring tied to increased social attention (*Dal Monte et al., 2022*). We quantified this by the distance between the centroid of the partner's face frame and the point of intersection of the gaze cone with the partner's facial plane (*Figure 4G*, left). Unfamiliar marmosets (both males and females) showed significantly higher (Mann-Whitney U test, unfamiliar male vs familiar male, p<0.0001; unfamiliar female vs familiar female, p<0.001) incidences of gaze toward the surrounding region of the partner's face (*Figure 4G*, right; compare darker lines with the lighter lines). Further, males exhibited markedly higher (Mann-Whitney U test, p<0.0001) incidences of gaze toward the partner females' face (*Figure 4G*, right; compare cyan lines with the orange lines).

Finally, our analysis of joint gaze distribution underscores that familiar pairs establish shared attention more flexibly across their social distance, whereas shared attention in unfamiliar pairs is constrained by spatial proximity. Joint gaze is crucial in primates as it underpins social communication and coordination, serving as a foundation for more complex behaviors like cooperation and shared attention (*Emery, 2000*; *Tomasello et al., 2005*). We analyzed joint gaze behaviors between marmoset dyads, first projecting the locations of joint gazes onto different 2D planes around the arena (*Figure 5A*). Significant asymmetry was observed in the distribution of joint gazes on the 'XY' and 'XZ' planes, with the majority of joint gazes occurring within the female's arena, regardless of familiarity. This is consistent with our earlier results, showing more

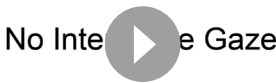

**Video 3.** Categorization of gaze behavior epochs of two freely viewing marmosets and transitions between the defined gaze states.

https://elifesciences.org/articles/105034/figures#video3

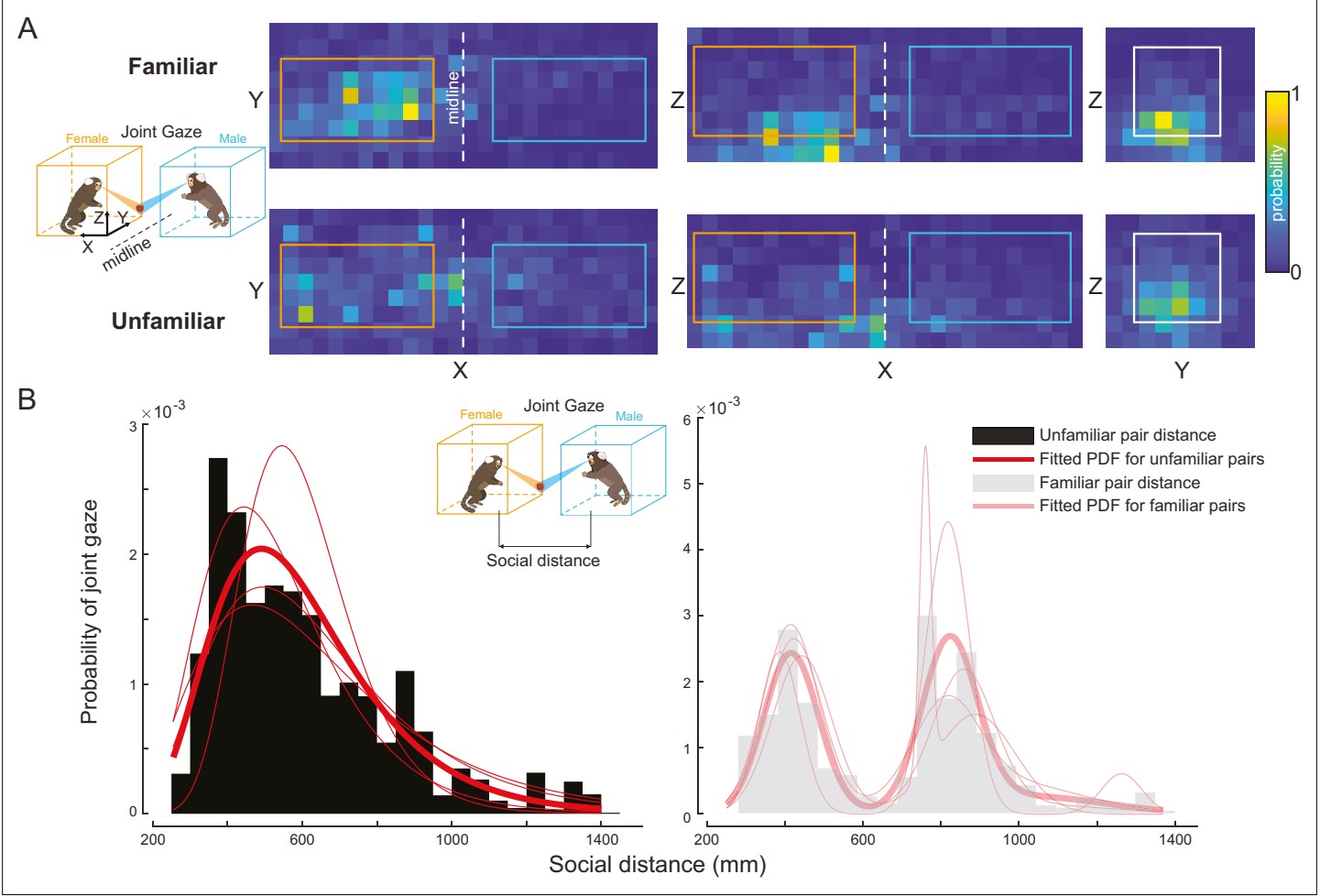

**Figure 5.** Joint gaze analysis between familiar and unfamiliar dyads. (**A**) Heatmaps of joint gaze locations projected onto two-dimensional (2D) planes (from left to right: '*XY*', '*XZ*', '*YZ*'). The icon on the left shows a schematic of a joint gaze along with the arena configuration with corresponding colors (orange for females and cyan for males). The colored rectangles superimposed on the heatmaps show the projections of the arenas onto those 2D planes. The white dotted line indicates the midline between the two arenas. (**B**) The probability of joint gaze at varying social distances shows distinct distributions for unfamiliar pairs (black) and familiar pairs (gray). The fitted red curve for the unfamiliar pairs is a lognormal distribution, while the fitted red curve for the familiar pairs is a tri-Gaussian distribution. Thin fitted red curves in both panels are the data from individual pairs. The inset shows a schematic of a joint gaze and the social distance metric. Social distance is defined as the Euclidean distance between the two marmosets.

partner-directed gazes from males and increased attention to the region surrounding the female partner (*Figure 4B, C, and G*). Notably, joint gazes between familiar pairs were more centrally distributed compared to those between unfamiliar pairs. When examining the '*XZ*' and '*YZ*' planes, we found that most joint gazes were concentrated along the lower *Z* axis, indicating that marmoset pairs tend to establish shared attention closer to ground level. Given the importance of social distance in shaping marmoset behaviors, we further investigated the distribution of joint gazes as a function of social distance. Striking differences were found between familiar and unfamiliar pairs (*Figure 5B*). The distribution for familiar pairs was best fitted by a tri-Gaussian model, consistent with the distribution of social distances observed in *Figure 3C*, suggesting that familiar pairs can establish joint gazes regardless of distance. In contrast, the distribution for unfamiliar pairs was best described by a lognormal model, indicating that unfamiliar pairs tend to establish joint gazes only when in close proximity.

Overall, we found that both the social dimensions we examined – familiarity and sex – are significant determinants of natural gaze dynamics among marmosets.

## Discussion

In this study, we first presented a novel framework for the automated, markerless, and identity-preserving tracking of 3D facial features of multiple marmosets. By building on top of the deep-learning framework provided by DeepLabCut, we used a constellation of cameras to overcome 'blind spots' due to occlusion and imposed spatiotemporal smoothness constraints on the detected features to establish and preserve identities across time. The tracked facial features from each animal form a semi-rigid face frame as the animal moves freely in 3D space, thereby allowing us to infer the animal's gaze direction at each moment in time. It is important to reiterate that unrestrained marmosets use rapid saccadic head movements for reorienting (*Pandey et al., 2020*) and have a limited amplitude range of saccadic eye movements (*Mitchell et al., 2014*). Thus, their head direction serves as an excellent proxy for gaze direction in unrestrained conditions.

Current methods for tracking and estimating animal body postures, such as DeepLabCut (*Mathis et al., 2018*) and SLEAP (*Pereira et al., 2022*), offer substantial advantages in precision and flexibility, particularly through the use of deep learning for markerless tracking across diverse species. These methods excel in laboratory settings, where they can achieve high accuracy without invasive markers and are adaptable for various animals and tasks. However, many of these tools struggle with occlusion, especially in crowded environments or natural habitats, limiting their effectiveness in complex social interactions or open-field studies (*Sturman et al., 2020*). Additionally, when tracking marmosets that exhibit significant vertical movement, requiring 3D tracking, the performance of these methods is suboptimal. While 3D tracking systems like DANNCE (*Dunn et al., 2021*) offer enhanced spatial accuracy, these methods are limited to single-animal tracking, which is insufficient for studies in social neuroscience.

Primates are highly visual species whose physical explorations of their environment are not confined to 3D surfaces. Gaze is a critical component of primate social behavior and conveys important social signals such as interest, attention, and emotion (*Emery, 2000*). Assessment of gaze is therefore important to understand non-verbal communication and interpersonal dynamics. Our 3D gaze tracking approach was able to capture both the positional and gaze dynamics of freely moving marmoset dyads in a naturalistic context. We observed clear effects of sex and familiarity on both interpersonal and gaze dynamics. Unfamiliar pairs exhibited more stereotyped patterns of arena occupancy, more sustained levels of social gaze across distance, and increased gaze monitoring, suggesting elevated levels of social attention and the need to constantly track the movements of a novel conspecific compared to familiar pairs. On the other hand, familiar pairs exhibited more recurrent joint gazes in the shared environment compared to unfamiliar pairs. Critically, familiar males also showed a higher tendency to reciprocate their partner's gaze, suggesting a greater awareness of their partner's social gaze state and a transition from mere monitoring to active social engagement.

Supported by the natural ethology of marmosets (*Yamamoto et al., 2014*; *Solomon and French, 1997*), we found dramatic sex differences in gaze behaviors, with males exhibiting significantly elevated levels of gaze toward females' faces and the surrounding regions, irrespective of familiarity. It is important to note that dominance in marmosets is not strictly determined by gender, as it can vary based on individual personalities and intra-group social dynamics, although breeding females typically dominate social activity within a group (*Digby, 1995*; *Mustoe, 2023*). While we have not explicitly controlled for dominance in this study, whether part of the observed differences can be attributed to dominance effects needs further exploration.

The distinction between social monitoring and social coordination is further reflected in the spatial flexibility of joint attention. Gaze following plays a crucial role in social communication for humans and nonhuman primates, allowing for joint attention (*Emery et al., 1997*; *Brooks and Meltzoff, 2005*; *Burkart and Heschl, 2006*; *Shepherd, 2010*). Previous research demonstrated that head-restrained marmosets exhibited preferential gazing toward marmoset face stimuli observed by a conspecific in a quasi-reflexive manner during a free-choice task (*Spadacenta et al., 2019*). Here, we found that males consistently followed the gaze of females into states of joint attention. This 'gaze following' occurred with similar frequency in both familiar and unfamiliar pairs, suggesting that monitoring a partner's gaze to coordinate shared attention is a fundamental component of marmoset social ethology, regardless of the strength of the social bond. Interestingly, we did not find any differences in gaze-following behaviors (transition from partner gaze to joint gaze) along the social dimensions we tested here. Future investigation of such behaviors by manipulating social variables such as dominance or kinship

could provide a comprehensive understanding of gaze following and joint attention in naturalistic behavioral contexts. The scarcity of reciprocal gazes in our study may be attributed to the task-free experimental setup employed. Indeed, in other joint action tasks requiring cooperation for rewards, marmosets actively engage in reciprocal gaze behaviors (*Miss and Burkart, 2018*).

While we focused on the tracking of facial features in this study, our automated system has the potential to extend to 3D whole-body tracking, encompassing limbs, tail, and the main body features of marmosets. In our system, multiple cameras surrounding the arena ensure that each body part of interest can be tracked through at least two cameras, enabling triangulation in 3D space. Our current system uses a pre-trained ResNet model (*He et al., 2016*) to track body parts of interest. However, considering the challenges posed by whole-body tracking, such as interference from marmosets' fur that complicates feature detection, the adoption of cutting-edge transformer networks like the vision transformer model (*Dosovitskiy, 2020*) might significantly improve detection performance. Such an advancement in tracking and reconstructing the entire marmoset body frame would enable the analysis of such data using unsupervised learning techniques (*Berman et al., 2014*; *Calhoun et al., 2019*) and thereby provide a deeper understanding of primate social behavior.

In summary, our study lays the groundwork for a rigorous quantification of primate behaviors in naturalistic settings. Not only does this allow us to gain deeper insights beyond what is possible from field notes and observational studies, but it is also a key first step to go beyond current reductionist paradigms and understanding the neural dynamics underlying natural behaviors (*Miller et al., 2022*).

## Methods

### Camera calibration

All cameras (GoPro 8) were calibrated using an 8-by-9 black-white checkerboard. For each camera, the checkerboard was placed at various locations to sample the space of the camera's field of view. To achieve better calibration performance, the checkerboard was tilted and rotated to varying degrees, thus producing a range of different views (*Zhang, 2000*). The corners of the checkerboard were automatically detected via a standard algorithm (detectCheckerboardPoints() function in the Image Processing and Computer Vision toolbox in MATLAB). The intrinsic parameters of each camera were estimated based on the data obtained from the checkerboard corner detection algorithm (estimateCameraParameters() function in Image Processing and Computer Vision toolbox in MATLAB).

### L-frame analysis

L-shaped frames were used to obtain the extrinsic parameters of the cameras, the rotation matrix, and the transition vector (*Dunn et al., 2021*). The L-shaped frame was captured by four cameras that recorded one arena. Four points that were unevenly distributed on the L-shaped frame were manually labeled. The information of transformation from world coordinates to camera coordinates was then extracted based on the labeled result (cameraPoseToExtrinsics() function in Image Processing and Computer Vision toolbox in MATLAB).

### Camera recording

GoPro 8 cameras were used and were simultaneously controlled via a Bluetooth remote control (The Remote by GoPro). Videos were recorded at 30 frames/s with a linear lens. Frame resolution was set at 1920×1080 pixels. A circular polarizer filter was used to mitigate reflection artifacts.

### Deep convolutional neural network model training

We used a modified version of DeepLabCut (*Mathis et al., 2018*) to perform automated markerless tracking of body parts of interest from two marmosets. The model was trained on 700 hand-labeled image frames extracted from videos of animals in their colony settings. Each image frame was labeled with six facial points: the two tufts, the central blaze, two eyes, and the mouth. The model was trained using GPUs on a large computing cluster for 250,000 iterations until the loss reached a plateau.

### Method validation in both 2D and 3D spaces

To validate the tracking method in 2D space, we manually labeled 200 continuous frames containing two marmosets from the recorded videos. The model was then provided with these 200 frames to

detect the body parts. We calculated the differences between the ground-truth coordinates and the detected body parts using the root mean square error (RMSE):

$$RMSE = \sqrt{\frac{\sum_{i=1}^{N} \left(x_i - \hat{x}_i\right)^2}{N}}$$

where $x_i$ is the ground-truth coordinate, $x_i$ is the detected coordinate, $N$ is the total number of body parts.

It is important to note that in cases where certain body parts were occluded in the frames, their coordinates were labeled as (0, 0).

To test the method in 3D space, we manually labeled 200 consecutive pairs of frames containing two marmosets, extracted from two simultaneously recording cameras. The coordinates were triangulated into 3D space. We then fed the model with these 200 pairs of frames to obtain the detected body part coordinates in 3D. The differences between the ground-truth coordinates and the detected body parts were calculated using the same RMSE metric.

### Gaze cone calculation

At each time frame, the gaze direction was calculated as the normal to the facial plane ('face norm vector') defined by the two eyes and the central blaze. The position of the ear tufts, which were behind this facial plane, was used to determine the direction of gaze. A gaze cone was defined as a virtual cone of 10-degree solid angle around this norm.

### Head gaze velocity calculation and stable epoch identification

We used the change of the norm over consecutive time frames to calculate the head gaze velocity:

$$v\left(t\right) = \frac{N\left(t+2\right) + N\left(t+1\right) - N\left(t-1\right) - N\left(t-2\right)}{6}$$

where $v(t)$ is the velocity at time point $t$, $N(t)$ is the face norm vector at time point $t$.

We remove all time points where the head gaze velocity was larger than 0.05 in normalized units. Segments no shorter than three consecutive time frames were identified as stable epochs.

### Cone-monitor plane intersection

We modified an existing method (*Calinon and Billard, 2006*; *Calinon, 2009*) to determine the elliptical intersection of a gaze cone and the finite plane defined by the monitor.

### Cone-facial plane intersection

We used a numerical method to determine whether the gaze cone of one animal intersected with the facial plane of the other. The facial plane was defined as the finite triangular plane formed by three facial features: two eyes and mouth. Any point $X$ in 3D within the volume bounded by the cone satisfies the inequality:

$$\cos\theta - \frac{dot\left(coneDir, X - coneOrg\right)}{norm\left(X - coneOrg\right)} \leq 0$$

where $\theta$ is the solid angle of the gaze cone, *coneDir* is the direction vector of the gaze cone, and *coneOrg* is the origin point of the gaze cone. The facial plane intersects with the cone if any point within the finite plane satisfies the inequality.

### Cone-cone intersection

To calculate the cone-cone intersection, we used the same numerical method as above. If any point $X$ in 3D simultaneously satisfied the following inequalities:

$$\cos\theta_1 - \frac{dot\left(coneDir_1, X - coneOrg_1\right)}{norm\left(X - coneOrg_1\right)} \leq 0$$

and

$$\cos \theta_2 - \frac{dot\left(coneDir_2, X - coneOrg_2\right)}{norm\left(X - coneOrg_2\right)} \leq 0$$

then the two cones were considered to be intersected. Subscripts in the above inequalities indicate the parameters of the two gaze cones under consideration.

### Gaze-type definition

To resolve spatial ambiguities where gaze cones might intersect both a partner and an external object, we implemented a hierarchical priority rule. A frame was categorized as Partner Gaze (or Reciprocal Gaze) if the gaze cone intersected the partner's facial plane, regardless of whether a secondary intersection with the other animal's gaze cone occurred elsewhere. Joint Gaze was only recorded when both animals' gaze cones intersected at an external point without either cone containing the partner's face. This hierarchy ensures that direct social attention is not masked by incidental external intersections.

### Maximum likelihood estimation

We used the maximum likelihood estimation method (mle() function in the Statistics and Machine Learning Toolbox in MATLAB) to fit a mixture of Gamma and Gaussian distributions.

### Markov chain analysis

State transition matrices were obtained based on the behavioral data. These matrices were then used to generate the discrete Markov chains (dtmc() function in Econometrics Toolbox in MATLAB) and plotted (graphplot() function in MATLAB).

### Warren Sarle's BC

Sarle's BC is used to test for bimodality. The coefficient is calculated using the MATLAB function written by *Zhivomirov, 2026*. The code is based on the theory in *Pfister et al., 2013*.

### Experimental model and subject details

#### Animals

Nine adult marmosets were used in this study (four males, five females)[Add info about age ranges of each sex]. Four familiar male/female pairs were each from the same cage. Four unfamiliar male/female pairs were selected from the nine animals such that each member of a pair was from different home cages and did not have visual access to each other while in the colony. Animals were kept in a colony maintained at around 75°F, 60% humidity, and a 12 hr:12 hr light-dark cycle. All procedures were approved by the Yale Institutional Animal Care and Use Committee (Yale University IACUC Protocol #2023-20163) and complied with the National Institutes of Health Guide for the Care and Use of Laboratory Animals.

#### Single marmoset gazing at the monitor

A single freely moving marmoset was recorded by four cameras surrounding the arena. Video or image stimuli were displayed at one of five locations (Center, Up, Down, Left, and Right) on the monitor (location chosen randomly). Each session contained only one stimulus category (either video or image) and consisted of five blocks. Each block consisted of ten 5 s stimuli interleaved with ten 5 s breaks. Each block started with a white dot in the center of the screen on a black background lasting for 1 s. At the end of the block, a juice reward (diluted condensed milk, condensed milk:water = 1:7) was delivered with a syringe pump system (NE-500 programmable OEM syringe pump from Pump Systems Inc) along with an auditory cue.

#### Freely interacting marmoset dyads

Two freely moving marmosets, in separate arenas, were recorded by two sets of four cameras surrounding the arenas. Each session consisted of ten 5 min free-viewing blocks interleaved with nine 5 min breaks. A juice reward (diluted condensed milk, condensed milk:water = 1:7) was delivered

every minute through two syringe pump systems during the free-viewing blocks. During the breaks, a divider was placed between the two arenas that prevented the marmosets from seeing each other.

## Acknowledgements

This research was supported by the National Institute of Mental Health (R21 120672, SWCC, ASN, MPJ; RF1 138396, SWCC, ASN, MPJ), Simons Foundation Autism Research Initiative (SFARI 875855, SWCC, ASN, MPJ), Brain Research Foundation (BRFSG-2020-05, ASN), Yale Orthwein Scholar Funds (ASN), and by the National Eye Institute core grant for vision research (P30 EY026878 to Yale University). We would like to thank the veterinary and husbandry staff at Yale for excellent animal care. We would like to thank Weikang Shi for the helpful discussion on the manuscript.

## Additional information

### Funding

| Funder | Grant reference number | Author |
|---|---|---|
| National Institutes of Health | R21 120672 | Monika P Jadi Steve WC Chang Anirvan S Nandy |
| Simons Foundation Autism Research Initiative | SFARI 875855 | Monika P Jadi Steve WC Chang Anirvan S Nandy |
| Brain Research Foundation | BRFSG-2020-05 | Anirvan S Nandy |
| National Institutes of Health | P30 EY026878 | Anirvan S Nandy |
| National Institutes of Health | RF1 138396 | Monika P Jadi Steve WC Chang Anirvan S Nandy |

The funders had no role in study design, data collection and interpretation, or the decision to submit the work for publication.

### Author contributions

Feng Xing, Data curation, Formal analysis, Investigation, Visualization, Methodology, Writing - original draft; Alec G Sheffield, Resources; Monika P Jadi, Anirvan S Nandy, Conceptualization, Supervision, Funding acquisition, Investigation, Writing – review and editing; Steve WC Chang, Conceptualization, Supervision, Funding acquisition, Writing – review and editing

### Author ORCIDs

Monika P Jadi ![ORCID] https://orcid.org/0000-0003-1092-5026
Steve WC Chang ![ORCID] https://orcid.org/0000-0003-4160-7549
Anirvan S Nandy ![ORCID] https://orcid.org/0000-0002-4225-5349

### Ethics

All procedures were approved by the Yale Institutional Animal Care and Use Committee (Yale University IACUC Protocol #2023-20163) and complied with the National Institutes of Health Guide for the Care and Use of Laboratory Animals.

Reviewer #1 (Public review): https://doi.org/10.7554/eLife.105034.3.sa1
Reviewer #2 (Public review): https://doi.org/10.7554/eLife.105034.3.sa2
Author response https://doi.org/10.7554/eLife.105034.3.sa3

## Additional files

**Supplementary files**
MDAR checklist

**Data availability**
Data and custom code used in this study have been deposited to Zenodo: https://doi.org/10.5281/zenodo.18928818.

The following previously published dataset was used:

| Author(s) | Year | Dataset title | Dataset URL | Database and Identifier |
|---|---|---|---|---|
| Xing F, Sheffield AG, Jadi MP, Chang SWC, Nandy AS | 2026 | Dynamic modulation of social gaze by sex and familiarity in marmoset dyads | https://doi.org/10.5281/zenodo.18928818 | Zenodo, 10.5281/zenodo.18928818 |

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
