## [Editor Report · eLife Assessment]

This study establishes a methodology (machine vision and gaze pose estimation) and behavioral apparatus for examining social interactions between pairs of marmoset monkeys. It has been difficult to study social interactions using artificial stimuli rather than genuine interactions between unrestrained animals. This study makes a **fundamental** contribution to social neuroscience research in a laboratory setting. Their results are **convincing** showing that the study of unrestrained social interactions is possible with detailed quantification of position and gaze. The methodology presented here is relevant to research in social neuroscience, neuroethology, and primatology.

---

## [Referee Report · Reviewer #1 (Public review)]

Summary:

The current study by Xing et al. establishes the methodology (machine vision and gaze pose estimation) and behavioral apparatus for examining social interactions between pairs of marmoset monkeys. Their results enable unrestrained social interactions under more rigorous conditions with detailed quantification of position and gaze. It has been difficult to study social interactions using artificial stimuli, as opposed to genuine interactions between unrestrained animals. This study makes an important contribution for studying social neuroscience within a laboratory setting that will be valuable to the field.

Strengths:

Marmosets are an ideal species for studying primate social interactions due to their prosocial behavior and the ease of group housing within laboratory environments. They also predominantly orient their gaze through head-movements during social monitoring. Recent advances in machine vision pose estimation set the stage for estimating 3D gaze position in marmosets but requires additional innovation beyond DeepLabCut or equivalent methods. A six point facial frame is designed to accurately fit marmoset head gaze. A key assumption in the study is that head-gaze is a reliable indicator of the marmoset's gaze direction, which will also depend on the eye position. Overall, this assumption has been well supported by recent studies in head-free marmosets. Thus the current work introduces an important methodology for leveraging machine vision to track head-gaze and demonstrates its utility for use with interacting marmoset dyads as a first step in that study.

Comments on revisions:

I thank the authors for their careful revisions of the manuscript. It has addressed all of my comments.

One final suggestion would be to add a scale bar in Supplemental Figure 2A so the size of the video/image stimuli is clear (in cm of monitor size) and also to report a range for how far away was the marmoset in viewing these stimuli (in cm). This will enable calculation of the rough accuracy in visual degrees.

---

## [Referee Report · Reviewer #2 (Public review)]

Summary:

The manuscript describes novel technique development and experiments to track the social gaze of marmosets. The authors used video tracking of multiple cameras in pairs of marmoset to infer head orientation and gaze, and then studied gaze direction as a function of distance between animals, relationships, and social conditions/stimuli.

Strengths:

Overall the work is interesting and well done. It addresses an area of growing interest in animal social behavior, an area that has largely been dominated by research in rodents and other non-primate species. In particular, this work addresses something that is uniquely primate (perhaps not unique, but not studied much in other laboratory model organisms), which is that primates, like humans, look at each other, and this gaze is an important social cue of their interactions. As such, the presented work is an important advance and addition to the literature that will allow more sophisticated quantification of animal behaviors. I am particularly enthusiastic about how the authors approach the cone of uncertainty in gaze, which can be both due to some error in head orientation measurements as well as variable eye position

Weaknesses:

While there remains some degree of uncertainty in the precise accuracy of the gaze measure, the authors have done an excellent job accounting for these as well as they can, and appropriately acknowledge the limitations of their approach.

Comments on revisions:

I have no further recommendations. The authors addressed my previous suggestions or acknowledged them as topics for future investigation. This is excellent work.

---

## [Author Response]

The following is the authors’ response to the original reviews

**Public Reviews:**

**Reviewer #1 (Public review):**
Summary:The current study by Xing et al. establishes the methodology (machine vision and gaze pose estimation) and behavioral apparatus for examining social interactions between pairs of marmoset monkeys. Their results enable unrestrained social interactions under more rigorous conditions with detailed quantification of position and gaze. It has been difficult to study social interactions using artificial stimuli, as opposed to genuine interactions between unrestrained animals. This study makes an important contribution for studying social neuroscience within a laboratory setting that will be valuable to the field.Strengths:Marmosets are an ideal species for studying primate social interactions due to their prosocial behavior and the ease of group housing within laboratory environments. They also predominantly orient their gaze through head movements during social monitoring. Recent advances in machine vision pose estimation set the stage for estimating 3D gaze position in marmosets but require additional innovation beyond DeepLabCut or equivalent methods. A six-point facial frame is designed to accurately fit marmoset head gaze. A key assumption in the study is that head gaze is a reliable indicator of the marmoset's gaze direction, which will also depend on the eye position. Overall, this assumption has been well supported by recent studies in head-free marmosets. Thus the current work introduces an important methodology for leveraging machine vision to track head gaze and demonstrates its utility for use with interacting marmoset dyads as a first step in that study.Weaknesses:One weakness that should be easily addressed is that no data is provided to directly assess how accurate the estimated head gaze is based on calibrations of the animals, for example, when they are looking at discrete locations like faces or video on a monitor. This would be useful to get an upper bound on how accurate the 3D gaze vector is estimated to be, for planned use in other studies. Although the accuracy appears sufficient for the current results, it would be difficult to know if it could be applied in other contexts where more precision might be necessary.

Please see our detailed responses to the reviewer comments below.

**Reviewer #2 (Public review):**
Summary:The manuscript describes novel technique development and experiments to track the social gaze of marmosets. The authors used video tracking of multiple cameras in pairs of marmosets to infer head orientation and gaze and then studied gaze direction as a function of distance between animals, relationships, and social conditions/stimuli.Strengths:Overall the work is interesting and well done. It addresses an area of growing interest in animal social behavior, an area that has largely been dominated by research in rodents and other non-primate species. In particular, this work addresses something that is uniquely primate (perhaps not unique, but not studied much in other laboratory model organisms), which is that primates, like humans, look at each other, and this gaze is an important social cue of their interactions. As such, the presented work is an important advance and addition to the literature that will allow more sophisticated quantification of animal behaviors. I am particularly enthusiastic with how the authors approach the cone of uncertainty in gaze, which can be both due to some error in head orientation measurements as well as variable eye position.Weaknesses:There are a few technical points in need of clarification, both in terms of the robustness of the gaze estimate, and possible confounds by gaze to non-face targets which may have relevance but are not discussed. These are relatively minor, and more suggestions than anything else.

Please see our detailed responses to the reviewer comments below.

**Reviewer #1 (Recommendations for the authors):**
Major comments:(1) It appears that the accuracy of the estimated gaze angle must be well under the size of the gaze cone (+/- 10 degrees), but I can't find any direct estimate of the accuracy even if it is just a ballpark figure. On Lines 219-233 is where performance is described for viewing images and video on a monitor, where it should be possible to reconstruct the point of gaze on the monitor while images and video are shown, in order to evaluate the accuracy of the system for where the marmoset is looking? Would you see eye position traces that would show fixation clusters around those images or videos with stationary points on the monitor much like that seen for head-fixed animals looking at faces on a screen (Mitchell et al, 2014)? If so, what is the typical spread of those clusters during fixations on an image, both in terms of the precision by RMS error during a fixation epoch and the spread around the images at different locations (accuracy of projection)? For example, if gaze clusters were always above the displayed images one would have an idea that the face plane is slightly offset above the true gaze direction. It is not completely clear how well the face plane and corresponding gaze cone do in describing gaze direction in space, but the monitor stimuli could be used as an initial validation of it.

We thank the reviewer for this important suggestion regarding the quantitative validation of gaze accuracy. We agree that, when animals view stimuli presented on a monitor, the estimated gaze direction can be evaluated by examining the spatial distribution of gaze–monitor intersection points relative to stimulus locations.

To address this, we generated a new figure (Fig. S2A) analyzing gaze behavior following the onset of video stimuli presented at different locations on the monitor. Specifically, we selected video clips in which human annotators verified that the marmosets were looking at the monitor. Consistent with prior work in head-fixed marmosets (Mitchell et al., 2014), we observe clustering of gaze–monitor intersection centers within and around the corresponding stimulus locations after stimulus onset. These clusters provide an empirical validation that the estimated gaze direction aligns with stimulus position in space.

Importantly, unlike the head-fixed preparation used in Mitchell et al. (2014), marmosets in our study were freely moving. As a result, they do not exhibit prolonged, stationary fixations on the monitor, and fixation clusters are therefore more diffuse. This increased spread reflects natural head and body motion rather than limitations of the gaze estimation method itself. Despite this, gaze intersection points remain spatially localized to the vicinity of the presented stimuli across different monitor locations.

We did observe small offsets in some gaze clusters relative to stimulus centers; however, these offsets were not systematic across stimulus locations or animals. Crucially, there was no consistent bias (e.g., clusters appearing uniformly above or below stimuli) that would indicate a systematic misalignment of the face plane or gaze cone relative to true gaze direction. Together, these observations support the conclusion that the face-plane-based gaze cone provides an accurate estimate of gaze direction in space, with precision well within the ±10° aperture of the gaze cone.

While the freely moving component of the behavior precludes direct estimation of fixation RMS error comparable to head-fixed paradigms, the observed stimulus-locked clustering serves as an initial validation of both the accuracy and practical utility of our approach under naturalistic conditions.

(2) A second major comment is about clarity in the writing of the results and discussion. At the end of the manuscript, a major takeaway is the difference between familiar and unfamiliar dyads, that males show more interest in viewing females including unfamiliar females, but for familiar females, this distinction is also associated with being likely to look at them if they look at the male, and then to engage in joint gaze with them after looking at them, which indicates more of a social interaction than simply monitoring them when they are unfamiliar. Those aspects of the results could be emphasized more in the topic sentences of paragraphs presenting data to support those features of the gaze data (at present is buried at the ends of results paragraphs and back in the discussion).

We thank the reviewer for this insightful suggestion. We have restructured the Results and Discussion sections to lead with the primary social takeaways rather than technical descriptions (Tracked changes in Word). Specifically, we now emphasize the distinction between "social monitoring" (characteristic of unfamiliar dyads) and "active social coordination" (characteristic of familiar dyads).

(1) Topic Sentences: We revised the topic sentences of all Results paragraphs to immediately highlight the findings regarding male interest and the influence of familiarity on reciprocation.

(2) Conceptual Framework: We added a conceptual distinction in the Discussion, explaining that while unfamiliar marmosets maintain high social attention through "peripheral monitoring" and proximity-dependent joint gaze, familiar pairs exhibit sophisticated, distance-independent coordination and gaze reciprocation.

(3) Clarification of Male Interest: We explicitly stated that while male interest in females is high regardless of familiarity, it manifests as persistent monitoring in unfamiliar pairs versus a more aware, reciprocal state in familiar pairs.

Minor comments:(1) Methods:a) Lines 522-539: The 200 continuous frames used for validation of the model containing two marmosets are sufficient to test how well it generalizes to other animals outside the training set? The RMSE reported, does it vary for animals inside vs outside the training set? To what extent does the RMSE, in image pixels, translate into accuracy in estimating the gaze direction, for example, as assessed by estimating error when marmosets look at images or video on the monitor?

To address the reviewer’s concern regarding generalization and the translation of pixel RMSE to angular accuracy, we emphasize that the six facial features selected are prominent, high-contrast features across the species. Consequently, we observed that the RMSE remained consistent for marmosets both inside and outside the training set. To quantify how pixel-level tracking error translates into gaze estimation accuracy, we performed a sensitivity analysis. We simulated landmark (i.e., feature) jitter by sampling perturbations from circular distributions based on our empirical data (2.4 pixels for eyes; 2.1 pixels for the central blaze). Our results, illustrated in uthpr response image 1, show that 90% of the resulting head gaze deviations fall within 10°, which is consistent with the angular threshold used for our gaze cone model. This confirms that the reported RMSE provides sufficient precision for reliable gaze estimation.

**Author response image 1. sa3fig1:** Probability distribution of gaze angular deviation under circular perturbation. The histogram (blue) represents the change in reconstructed gaze angle (degrees) following stochastic perturbation of facial features. To simulate real-world variance, noise was sampled from circular distributions with radii of 2.4 pixels (eyes) and 2.1 pixels (central blaze). The red curve represents an exponential fit to the empirical data (y=ae^bx^, a=0.9591, b=0.1813). Approximately 90% of the reconstructed gaze deviations remain below 10°, indicating the model’s localised stability under pixel level coordinate jitter.

b) Line 542-43: Is there any difference between a rigid model fit to the six facial points, versus using the plane defined by the two eyes and central blaze in terms of direction accuracy (in the ground truth validation)? How does the "semi-rigid" set of six points (mentioned also in lines 201-203) constrain the fit of the three points (two eyes and central blaze) that define the normal plan for the gaze cone?

We thank the reviewer for the opportunity to clarify our geometric model. The plane used to define the gaze cone's origin was indeed determined by the two eyes and the central blaze. However, a plane defined by only three points was insufficient to determine a unique gaze direction, as the normal vector was ambiguous (it could point forward through the face or backward through the head).

To resolve this, we utilized the relative positions of the two ear tufts. Because the tufts are anatomically situated behind the eyes and blaze, these additional points provide the necessary spatial context to orient the gaze vector correctly. In our validation, we found that the mouth does not alter the angular accuracy compared to a 3-point fit, supporting that the facial features are correctly identified.

We use the term 'semi-rigid' to describe the six-point constellation because their relative spatial configurations remain stable across individuals and expressions, imposing a biological constraint on the model. This prevents unphysical warping of the face frame during 3D reconstruction and ensures the gaze cone remains anchored to the animal's true midline.

(2) Results:a) Lines 203-205: What is the distinction between gaze orientation (defined by facial plane, 3D vector) and gaze direction (defined by ear tufts) ... is gaze direction in the 2D x-y plane? Why are two measures needed or different? It does not appear gaze orientation is used further in the manuscript and perhaps could be omitted.

We appreciate the reviewer’s comment regarding the terminology. We have replaced all instances of ‘gaze orientation’ with ‘gaze direction’ to ensure consistency throughout the manuscript.

To clarify, both terms referred to the same 3D unit vector. The ear tufts were not used to define a separate 2D measure; rather, they served as posterior anatomical anchors to resolve the 3D polarity of the normal vector (ensuring the vector points 'forward' from the face rather than 'backward'). Gaze direction was calculated in 3D space and was not restricted to a 2D x-y plane. We have clarified this in the revised Methods section (Lines 203–205) to avoid further ambiguity.

b) Line 215-216: why is head-gaze velocity put in normalized units instead of degrees visual angle per second? How was the normalization performed (lines 549-557)? It would be simpler to see velocity as an angular speed (degrees angle per second) rather than a change in norms.

We thank the reviewer for this suggestion. We agree that the expression is misleading.

(1) We have replaced "face norm" with "face normal vector" (N) throughout the manuscript to clarify that we are referring to the 3D unit vector perpendicular to the facial plane.

(2) Lines 224-225 and the corresponding Methods section (Lines 599-609) have been updated to reflect this change in units and terminology.

We chose to use the change in the face normal vector in normalized units for our primary calculations because it allows for efficient spatiotemporal smoothing and is computationally robust at the very low thresholds required for our stability analysis. However, to address the reviewer's concern regarding interpretability, we have verified that our threshold of 0.05 normalized units corresponds to an angular velocity of 2.87 degrees/frame duration [33ms]. Since we are operating at very small angular changes, the Euclidean distance between unit vectors is a near-linear proxy for the angular displacement in radians.

c) Lines 215-216: How do raw gaze traces appear over time ... are there gaze saccades and then stable fixations, or does it vary continuously? A plot of the gaze trace might be useful besides just showing velocity with a threshold, to evaluate to what extent stable fixation vs shifts are distinct.

**Author response image 2. sa3fig2:** Time course of gaze, angular velocity and stability, thresholding. The plot illustrates the temporal dynamics of the face normal vector velocity used to define stable gaze states. The blue trace represents the raw gaze velocity calculated in normalised units. The red dashed line demotes the empirical cut off threshold of 0.05 units per frame.

To clarify the temporal dynamics of marmoset head movements, we have provided a representative time course of head gaze velocity as shown in Author response image 2. The data clearly show a "saccade-and-fixate" pattern: large, distinct spikes in velocity (representing rapid head redirections) are separated by periods of relative stability.

While minor high-frequency fluctuations in the raw trace (blue) may be attributed to facial feature detection noise, they remain significantly below our stability threshold (red dashed line). By applying this threshold, we successfully isolated biologically relevant "stable fixations" from "head saccades," ensuring that our subsequent social gaze analysis is based on periods of intentional head gaze direction.

d) Lines 237-286: The writing in this section does not emphasize the main results. There seem to be three takeaway points that could be emphasized better in the topic sentences of each of the paragraphs: i) Marmosets tended to spend most of their time on either end of the elongated box, not in the middle, ii) Males spent more time near the front of the box near the other animal than females, iii) Familiar pairs spent more time closer to each other.

To address this comment, we have reorganized this section to lead with the three key behavioral findings:

(1) We now state clearly in the topic sentence that marmosets preferred the ends of the arena over the middle.

(2) We have highlighted the finding that males spend significantly more time near the inner edge (closer to the partner) than females, irrespective of familiarity.

(3) We emphasized that familiar pairs maintain closer and more dynamic social distances over time, whereas unfamiliar pairs tend to move further apart as a session progresses.

e) Line 303: It would be useful to see time traces of head velocity of each member of the pair and categorization over time of the gaze event types. A stable epoch must be brief on the order of 100-200ms. It is unclear how distinct the stable fixation epochs are from the moments when the gaze is shifting. Also, the state transition analysis treats each stable epoch like one event, and then following a gaze movement by either of the pair, the state is defined again, is that correct?

We defined stable epochs as continuous periods where the face normal vector velocity remained below 0.05 normalized units for both animals. This ensures that a "gaze state" is only categorized when both marmosets have relatively fixed head orientations. As shown in the provided time traces in Author response image 2, the velocity profile is characterized by sharp peaks (head saccades) and clearly defined troughs (fixations). Further, we generated a probability histogram of stable head-gaze epoch durations (Author response image 3). The median duration of these stable epochs is 200ms, which aligns with biological expectations for fixation durations in primates and confirms that these states are distinct from the high-velocity shifts.

The reviewer’s interpretation is correct. Our Markov chain model treats each stable epoch as a single event. A transition occurs when at least one animal moves (exceeding the velocity threshold), resulting in a new stable epoch where the relative gaze state is re-evaluated. This approach allows us to model the sequence of social interactions as a series of discrete behavioral decisions.

**Author response image 3. sa3fig3:** Temporal characteristics of stable gaze, head gaze, epochs. The histogram illustrates the probability distribution of the duration (ms) of stablegaze behaviour epochs. A minimum duration threshold of 100 ms was applied to exclude transient, non-purposeful head gazes.

f) Lines 316-326: Some general summarizing statements to lead this paragraph would be useful. It seems that familiar pairs are more likely to participate in joint gaze, especially when close to each other, and perhaps, that males tended to gaze at females more than the reverse. Is there any notion that males were following the gaze of females?

We thank the reviewer for these suggestions. We have revised the topic sentences of this section to lead with a summary of the social takeaways, specifically highlighting the higher level of male interest and the shift toward reciprocal coordination in familiar pairs.

The reviewer correctly identified an important dynamic. Our transition analysis (Fig. 4D) confirms that males in both familiar and unfamiliar dyads frequently follow the female's gaze. This is evidenced by a robust transition probability (~17%) from "Male-to-Female Partner Gaze" (blue node) to "Joint Gaze" (green node). We found that this gaze-following behavior was a general feature of the dyads and did not differ significantly by familiarity, which is why it was not previously emphasized. However, we have now added a statement to the Results (Lines 358-365) to explicitly describe this male-led gaze-following behavior.

g) Lines 328-337: Can these findings in this paragraph be summarized more generally? It seems males view unfamiliar females longer, whereas for familiar females they are more likely to reciprocate viewing if being viewed by them and then to join in joint gaze with them. Would that event, viewing a female and then a transition to joint gaze, not be categorized as a gaze-following event?

We have now summarized the paragraph to emphasize the transition from vigilant monitoring in unfamiliar pairs to reciprocal awareness in familiar pairs.

Regarding "longer" viewing: We have clarified the text to specify that males' interest in unfamiliar females is persistent and robust rather than simply "longer" in a single duration. The high recurrence probability signifies that males consistently re-orient their gaze back to the unfamiliar female even if the interaction is briefly interrupted by movement.

Regarding gaze following and joint gaze: The reviewer asks if the transition from viewing a female to joint gaze constitutes gaze following. We agree that a transition from "male-to-female gaze" to "joint gaze" is indeed a gaze-following event (as noted in our previous response regarding Fig. 4D). However, the specific transition discussed in this paragraph (female-to-male gaze to male-to-female gaze) is different: it describes a "reciprocal" event where the male responded to being looked at by looking back at the female, while the female simultaneously shifted her gaze away. Since the two gaze cones did not intersect on an external object or on each other's faces simultaneously at the end of this transition, it was not categorized as joint gaze or gaze following.

h) Lines 339-351: It is not clear why gazing at the region surrounding a female's face (as opposed to the face itself) reflects "gaze monitoring tied to increased social attention (Dal Monte et l, 2022). This hypothesis could be expanded to make the prediction clear in this paragraph.

We thank the reviewer for identifying the need to clarify the hypothesis regarding the region surrounding the face. We have expanded this paragraph to explain why gazing at the peripheral facial region reflects social monitoring.

In many primate species, direct and sustained eye contact can be often interpreted as a threat or a challenge, particularly between unfamiliar individuals. Peripheral monitoring (looking at the area immediately surrounding the face) can strategically allow an animal to stay highly attentive to the partner's head orientation, gaze direction, and facial expressions—all critical for anticipating future actions—while minimizing the risk of social conflict. By demonstrating that unfamiliar marmosets utilize this peripheral strategy significantly more than familiar ones, we provide evidence that social attention in novel dyads is characterized by a social monitoring strategy that balances the need for information with social caution.

i) Lines 354-373: This section seems to suggest again that in a familiar male/female pair, the male is more likely to follow the female gaze and establish a joint gaze, and this occurs less with the unfamiliar pair only when closer in distance. Some summary sentences to begin the paragraph could help frame what to expect from the results.

We have added summarizing topic sentences to this section to clarify the relationship between familiarity and the spatial distribution of joint gaze.

(3) Discussion:Lines 380-463: This section reads more clearly than most of the results, where it is often hard to connect the data plots to their significance for behavior. Overall, I believe the manuscript could be improved by setting up a hypothesis before presenting results in the paragraphs demonstrating the data. Some of the main findings appear in text from lines 413-419 (somewhat hidden even in discussion).

We sincerely appreciate the reviewer’s positive feedback on the clarity of the latter sections of our Discussion. We have taken the suggestion to heart and have performed a comprehensive restructuring of the Results and Discussion sections.

(1) We have moved the key takeaways, specifically the distinction between vigilant monitoring in unfamiliar pairs and reciprocal coordination in familiar pairs, from the end of the Discussion to the topic sentences of the relevant Results paragraphs.

(2) We established a unified framework throughout the manuscript that connects pixel-level tracking stability to the biological "saccade-and-fixate" movement pattern, and ultimately to the social dimensions of sex and familiarity.

(4) A couple of additional questions to address in the discussion:a) Can you speculate why in this behavioral context the marmosets do not engage in reciprocal gaze where both are simultaneously looking at each other (lines 297-301)? How low is the incidence of this event, numerically, in comparison to the other events (1 in 1000 events, etc)?

We appreciate the reviewer’s interest in the lack of reciprocal gaze (mutual eye contact).

Numerically, reciprocal gaze events occurred with a frequency of approximately 1 in 500 social gaze events (comprising less than 0.2% of our social dataset). Given this extreme scarcity, we felt that any statistical comparisons across sex or familiarity would be underpowered and potentially misleading, leading to our decision to focus on partner and joint gaze states.

We speculate that the rarity of reciprocal gaze is primarily due to our task-free experimental setup. Unlike directed cooperation tasks where animals must look at each other to coordinate actions for a reward (e.g., Miss & Burkart, 2018), our study focused on task-free interactions. In a free-moving context without a common goal, marmosets may prioritize monitoring the environment or the partner’s actions (joint or partner gaze) over direct, sustained mutual eye contact, which can sometimes be perceived as a confrontational or high-arousal signal in primate social hierarchies.

b) Does a transition from a marmoset viewing their partner, to a joint gaze, count as a gaze-following event? It appears the authors are reluctant to use that terminology. What are the potential concerns in that terminology? Is there a concern that both animals orient to the same object that is salient to them without it being due to their gaze?

A transition from a partner-directed gaze to a joint gaze is indeed a gaze-following event. We distinguish these events from a transition between partner-directed gazes (e.g., male-to-female to female-to-male). In these "reciprocation" cases, once the second animal looked at the first, the first animal shifted their gaze away. Because the two gaze cones did not intersect on a common object at the end of the transition, I classified such events as a social exchange of attention rather than a coordinated gaze-following event.

**Reviewer #2 (Recommendations for the authors):**
I do have a few questions/points for clarification:(1) While your approach appears to be able to track head orientation when the face is occluded or turned away from the primary cameras, how was the accuracy of this validated? Since you have multiple cameras, it should be possible to make the estimate using the occluded cameras and then validate using the non-occluded ones.

We appreciate the reviewer's comment regarding the validation of our tracking during partial occlusions.

We wish to clarify that our system does not utilize "primary" vs "auxiliary" cameras. Rather, any two or more cameras that capture facial features with high confidence are used to triangulate the points into 3D space. Thus, the "primary" cameras are dynamically determined frame-by-frame based on the animal's orientation.

To validate the accuracy of our 3D reconstruction during occlusions, we utilized a "projection-validation" approach. As demonstrated in Figure 2B (left panel), when the face is turned away from a specific camera, leaving only the back of the head visible, we used the facial features triangulated from the other non-occluded cameras and projected them onto the image plane of the occluded camera. The fact that these projected points aligned precisely with the expected (but hidden) anatomical landmarks confirms the global accuracy of our 3D model.

We previously benchmarked this approach using a three-camera system where we triangulated coordinates via two cameras and successfully projected them onto the third camera's image plane with high accuracy. This ensures that even when a camera is "blind" to the face, the 3D position estimated by the rest of the array remains robust.

(2) Marmosets, like other non-human primates, also look at other body postures for their social communication, though admittedly marmosets are far more likely to look others in the face than larger primates. The tail-raised genital displays come to mind. While the paper primarily focuses on shared vs deviant gaze, and I believe tracks not only the angle of viewing towards the target but also the distance from the face (please clarify if I am wrong), it would also be useful to know how often marmosets are looking at each other beyond just the face. This is particularly interesting if the gaze towards the partner varies depending on whether that partner was generally oriented towards the gazer, or not. For the joint gaze, were there conditions in which the two were looking at the same target, but had body postures that were not oriented toward one another (i.e. looking at a distant target beyond one of the animals, like looking over someone else's shoulder)?

We thank the reviewer for highlighting the importance of body postures and non-facial social signals (e.g., genital displays) in marmoset communication.

At the inception of this project, we explored tracking multiple body parts. However, due to the marmoset's dense fur and the lack of distinct skeletal markers under naturalistic lighting, human annotators and early automated tools struggled to achieve the precision required for high-resolution 3D kinematics. While recent advances in whole-body tracking now make these questions approachable, we chose to focus on the face normal vector because it provided the most robust and high-confidence signal for social orientation in our current dataset.

Regarding the "looking over the shoulder" scenario, we utilized a hierarchical classification system to prevent wrong categorization. Intersection with the partner’s face always took priority. If one animal’s gaze cone contained the other’s face, the state was classified as "Partner Gaze", even if the two gaze cones happened to intersect at a distant point in space. This ensures that "Joint Gaze" specifically captures instances where both animals ignore one another’s face regions to focus on a shared external target.

We agree that the relationship between body posture and head gaze is a fascinating area for future research. In our current setup, while "Joint Gaze" requires the head-gaze cones to intersect, the animals' bodies could indeed be oriented in different directions (e.g., looking at a distant target behind the partner). We have added a note to the Discussion acknowledging that incorporating whole-body gestures would further deepen the understanding of marmoset social ethology.

(3) In the introduction, (line 70), you raise the question of ecological relevance, using rhesus in laboratory settings. This could use a little more expansion/explanation of the limitations of current/past approaches.

We thank the reviewer for the suggestion to expand upon the ecological limitations of traditional laboratory paradigms.

We have substantially revised the Introduction (Lines 70–82) to provide a more detailed critique of past approaches. Specifically, we now highlight how traditional head-fixed or screen-based paradigms decouple eye movements from natural head-body dynamics and lack the reciprocal, multi-agent complexity found in real-world social environments (e.g., Land, 2006; Shepherd, 2010). By contrasting these constraints with the spatially and socially embedded nature of marmoset interactions, we clarify why a more naturalistic, quantitative approach is necessary to understand the true dynamics of social gaze. These additions provide a stronger theoretical foundation for our move toward a free-moving experimental model.